# Novel Breast Cancer Brain Metastasis Patient-Derived Orthotopic Xenograft Model for Preclinical Studies

**DOI:** 10.3390/cancers12020444

**Published:** 2020-02-14

**Authors:** Masanori Oshi, Maiko Okano, Aparna Maiti, Omar M. Rashid, Katsuharu Saito, Koji Kono, Ryusei Matsuyama, Itaru Endo, Kazuaki Takabe

**Affiliations:** 1Breast Surgery, Department of Surgical Oncology, Roswell Park Comprehensive Cancer Center, Buffalo, NY 14263, USA; masa1101oshi@gmail.com (M.O.); tentekomaikocco@hotmail.com (M.O.); Aparna.Maiti@RoswellPark.org (A.M.); 2Department of Gastroenterological Surgery, Yokohama City University School of Medicine, Yokohama 236-0004, Japan; ryusei@yokohama-cu.ac.jp (R.M.); endoit@med.yokohama-cu.ac.jp (I.E.); 3Department of Breast Surgery, Fukushima Medical University School of Medicine, Fukushima 960-1295, Japan; 4Department of Surgery, Holy Cross Hospital, Michael and Dianne Bienes Comprehensive Cancer Center, Fort Lauderdale, FL 33308, USA; omarmrashidmdjd@gmail.com; 5Department of Gastrointestinal Tract Surgery, Fukushima Medical University School of Medicine, Fukushima 960-1295, Japan; k-yamame@fmu.ac.jp (K.S.); kojikono@fmu.ac.jp (K.K.); 6Department of Surgery, Jacobs School of Medicine and Biomedical Sciences, State University of New York, Buffalo, NY 14263, USA; 7Department of Surgery, Niigata University Graduate School of Medical and Dental Sciences, Niigata 951-8510, Japan; 8Department of Breast Surgery and Oncology, Tokyo Medical University, Tokyo 160-8402, Japan

**Keywords:** brain metastasis, breast cancer, epothilone, gene expression, mouse model, orthotopic implantation, PDX, pre-clinical model, xenograft

## Abstract

The vast majority of mortality in breast cancer results from distant metastasis. Brain metastases occur in as many as 30% of patients with advanced breast cancer, and the 1-year survival rate of these patients is around 20%. Pre-clinical animal models that reliably reflect the biology of breast cancer brain metastasis are needed to develop and test new treatments for this deadly condition. The patient-derived xenograft (PDX) model maintains many features of a donor tumor, such as intra-tumor heterogeneity, and permits the testing of individualized treatments. However, the establishment of orthotopic PDXs of brain metastasis is procedurally difficult. We have developed a method for generating such PDXs with high tumor engraftment and growth rates. Here, we describe this method and identify variables that affect its outcomes. We also compare the brain-orthotopic PDXs with ectopic PDXs grown in mammary pads of mice, and show that the responsiveness of PDXs to chemotherapeutic reagents can be dramatically affected by the site that they are in.

## 1. Introduction 

Despite popularization of breast screening and improvements in treatment over the last two decades, more than 40,000 patients die from breast cancer annually in the United States [1]. Metastasis, distant dissemination of cancer, is the major cause of death in these patients. In particular, metastasis to the brain is the deadliest, with a dismal 20% one-year survival rate which occurs in nearly 30% of breast cancer patients [2]. One of the reasons for the poor prognosis of patients with brain metastasis is that there is no established treatment option other than cranial radiotherapy [3,4]. The development of novel therapeutics for this disease is clearly an urgent need. Several efforts have been made or are ongoing in this regard [5], but they have been hampered by the lack of preclinical models that reliably reproduce clinical characteristics of brain metastasis [6]. Animal models that do not reproduce patient conditions mislead the preclinical study results that consequently lead to the failure of clinical trials that not only waste resourced, but even worse, expose patients with ineffective interventions [7].

While genetically engineered mouse models of breast cancer have significantly contributed to the identification of the roles of specific genes in tumor development, they have a low incidence of brain metastasis and do not fully reflect the disease in humans [8,9]. A number of cell-lines have been developed for preclinical breast cancer brain metastasis research [10]. However, the clinical and biological heterogeneity of human tumors cannot be fully reproduced in cell-lines [11,12,13]. Though brain metastases in mice can be generated by directly injecting cells into the blood circulation through the tail vein or into the heart [14], such injections lead to systemic distribution of cells to other organs besides brain. The intracarotid artery injection method was developed to minimize the spread of cells to areas other than the brain [15]. However, this method requires microsurgery skills, and is complicated by a high post-operative mortality rate.

The patient-derived xenograft (PDX) model of cancer is created by transplanting tumor tissue in immunodeficient mice at the same (orthotopic) or different (ectopic) anatomic site. PDXs retain genetic, transcriptional and phenotypic features of the original tumor even after long-term continuous passage in vivo [16,17,18,19,20,21], and they have been used successfully to develop new therapies for cancers [22,23]. Subcutaneous breast cancer PDXs are often used for pre-clinical drug screening because both tumor transplantation and tumor tracking are easy. However, orthotopic PDXs have more biological information about the original human tumor than subcutaneous PDXs [24], and can better predict drug treatment effects [25,26]. This is expected because orthotopic PDXs retain the environmental features of the original tumor and the tumor microenvironment has a significant impact on the biological behavior of cancer [27,28,29].

Although several papers have been published on orthotopic implantation in mice of human brain metastatic lesions, the technique of tumor implantation in the mouse brain remains a surgically difficult procedure. We have developed a number of breast cancer mouse models and reported their usefulness for pre-clinical studies [7,30,31,32,33]. Here, we have established a novel orthotopic breast cancer brain metastasis PDX model that better mimics human cancer than the commonly used ectopic PDX. 

## 2. Results

### 2.1. Comparison of Three Tumor Implantation Methods to Generate Brain PDXs

To identify the most efficient technique for tumor implantation in mouse brain, we evaluated three methods: direct implantation of a tumor fragment using fine-point forceps (“Forceps” method), implantation of a gently crushed tumor fragment using beveled 23 G needle with syringe (“Needle” method), and implantation of a gently crushed tumor fragment using a 10-µL plastic tip with pipette (“Pipette” method). Brain metastasis tumor sample of a triple-negative breast cancer patient (Appendix A) that had been passaged three times in mouse brain was used. This was done so that adequate tumor tissue was available for the experiment to compare the implantation methods. The tumor was minced with a surgical blade into approximately 1 mm^3^ fragments (Figure 1A,B). These fragments were directly used for implantation with the Forceps method. For the other two methods, 2 µL of phosphate-buffered saline (PBS) was added to a fragment in a 0.5-mL tube and the fragment was gently crushed against the wall of the tube using a pipette tip (Figure 1B), and the entire 3 µL tumor preparation was inoculated. With all three methods, the tumor fragments were implanted in the caudate putamen at X, Y, and Z coordinates of respectively 2, 1, and 4 mm with respect to the bregma through a 2-mm wide burr hole in the skull (Figure 1C,D). For the Pipette method, the pipette tip was shortened by removing its distal 1 cm of a 10-µL plastic tip (Figure 1E). Of note, 1 cm from the shortened tip was marked to measure the exact depth to insert the tip. With all methods, the implantation procedure from anesthesia to closure of skull hole took 5 min per mouse.

Implantation by all methods were performed by the first author. The brains of 8–10 mice were implanted with each method. Post-operative mortality was zero with the Forceps and Pipette methods; however, five of the eight mice died within a day of implantation with the Needle method (Figure 2A). Tumor growth was monitored by MRI every other week (Figure 2A). Tumors were successfully engrafted in 80%, 66.6%, and 100% of mice at 8 weeks after implantation with the Forceps, Needle, and Pipette methods, respectively (Figure 2A). Furthermore, the volume of the tumors 6 weeks after implantation was greater with the Pipette and Forceps methods compared to the Needle method (Figure 2B). The superiority of the Pipette method was also observed in a separate experiment that used brain metastasis tumor of a different patient (Appendix A), whose primary breast cancer was of ER+ subtype. In this experiment too, post-operative mortality was zero with both the Pipette and Forceps methods (Appendix A). Tumor engraftment rate at 8 weeks was 100% and 75%, respectively, with the Pipette and Forceps methods. Given the high postoperative survival, tumor engraftment rate, and tumor size consistency, the Pipette method thus appeared to be the best to generate orthotopic PDXs of brain metastases.

### 2.2. Enzymatic Tumor Dissociation Generated Smaller Brain PDX Tumors

As noted above, our samples were gently crushed in PBS, whereas samples are commonly dissociated enzymatically before implantation. Therefore, we examined if the technique of tumor fragmentation, non-enzymatic vs. enzymatic dissociation, changes the efficiency of tumor implantation by the Pipette method. Brain metastasis of one breast cancer patient (Appendix A) that had been passaged three passages as brain PDX was used for this experiment. 2 µL of RPMI-1640 with collagenase and hyaluronidase at commonly used concentrations [34] were added to a tumor fragment (1 mm^3^) and the mix was incubated at 37 °C for an hour, after which the tumor preparation was washed and resuspended to 3 µL with PBS for Digestion method. Post-operative mortality was zero regardless of enzymatic digestion technique. However, re-transplantation rate at 8 weeks was only 33% for tumors that underwent digestion compared to the 100% rate for tumors that did not undergo digestion (Figure 3A). Furthermore, tumors that were digested before implantation grew to only one-fourth the size of tumors that were not digested (*t* test *p* = 0.032; Figure 3A).

### 2.3. Matrigel Did Not Promote Growth of Brain PDX Tumors

PDX tumors are commonly implanted within Matrigel extracellular matrix. To assess the effect of this matrix on tumor growth in the brain with the Pipette method, tumor fragments were implanted in the brain with 2 µL of PBS or Matrigel. All implanted mice had tumor growth. However, tumors implanted with Matrigel were about half the volume compared to PBS at 8 weeks after implantation (*t* test *p* < 0.05; Figure 3B). Median tumor doubling times, determined from tumor growth during 6th to 8th weeks, were approximately 4 and 6 days with PBS and Matrigel, respectively, and significantly different (*p* < 0.05).

### 2.4. Orthotopic Brain Metastasis PDX Engrafted Better But Grew Similarly Compared with Ectopic PDXs

Because of its technical ease, ectopic implantation of brain metastases in mammary fat pads (MFP) of mice is commonly performed in PDX models. We therefore compared the tumor growth of ectopic and orthotopic PDXs. Brain metastasis of one triple negative breast cancer patient (Appendix A) was used for this experiment. Tumor fragments (1 mm^3^) in 2 µL PBS were implanted in the brain with the pipette method or in mammary fat pads using standard methods (*n* = 3). The PDXs were passaged at the same site for three generations (Figure 4). The engraftment rate at 8 weeks was 100% for orthotopic implantation; however, it was only 75% for ectopic implantation. The engraftment rate improved to 87% after three passages (Figure 4B). The engraftment rate for brain PDXs was not affected by passaging (Figure 4A). However, tumor growth was about two times faster after three passages (*t* test *p* < 0.05). Such an increase in growth rate with passaging was also observed for brain metastatic lesions of two other patients (Appendix A). Like orthotopic PDXs, ectopic PDXs in MFP too had improved growth after passaging (*p* < 0.05; Figure 4B). Orthotopic and ectopic PDXs had similar growth rates at any given passage number.

### 2.5. Transcriptomic Profile of Orthotopic PDX was More Similar to the Original Brain Tumor than Ectopic PDX

To assess the similarity of the human tumor with its PDX grown in either mouse brain or mouse MFP, we compared transcriptomes of one patient’s brain metastasis (Appendix A) and its PDXs that had been concurrently passaged three times in mouse brain or MFP. Global gene expression profiles of these three samples were generated using RNA sequencing. To account for RNAs arising from host mouse cells infiltrating the PDXs, we utilized a bioinformatics workflow to filter out mouse RNAs from the sequencing data. Such RNAs were responsible for about 15% of sequencing reads of the PDX samples (Figure 5A). As expected, in the case of brain PDX, these mouse transcripts included those of genes such as *Dscam* (Down syndrome cell adhesion molecule) and *Gfap* (glial fibrillary acidic protein) that are highly expressed in brain cells. For the MFP PDX, the genes included *Adig* (adipogenin) and *Mmp9* (matrix metallopeptidase 9), which have high expression in adipocytes [35].

The transcriptomes of both the brain and MFP PDXs were different from the human brain metastasis (Figure 5B). This is expected because non-cancerous tumor stromal cells, such as endothelial and fibroblast cells, were represented in the transcriptome of the brain metastasis, whereas they were not in the case of the PDXs since transcripts arising from their stromal cells were excluded from the transcriptome data because stromal cells in PDXs are of mouse origin. We expected that the transcriptomes of the two PDXs would be similar. However, visual examination of the heatmap of global gene expression in these samples showed that the two samples were transcriptionally different (Figure 5B). The heatmap did not reveal which of the two PDXs was more similar to the patient’s brain metastasis. We therefore analyzed the gene expression at the level of gene sets because expression for sets of functionally related genes instead of individual genes can be more reflective of biology. We summarized the gene expression values of the samples into a smaller set of gene set scores that were calculated for Reactome gene sets using the single-sample gene set variation analysis (GSVA) method. Unsupervised clustering of the three samples using these scores indicated that between the two PDXs, the orthotopic PDX was more similar to the patient tumor for functional gene expression than the tumor’s ectopic PDX (Figure 5C).

### 2.6. Orthotopic and Ectopic Brain Metastasis PDXs Responded Differently to Drug Treatment

The aim of this study was to establish a novel brain metastasis PDX mice model for drug development. We expect it to assess the effectiveness of the test compound on metastatic brain tumor which is not always the same with the primary tumor prior to the compound entering clinical trials. Epothilone B has high anti-cancer activity against primary breast cancer and can cross the blood-brain barrier [36], thus, clinical trial was performed and failed due to its ineffectiveness against breast cancer brain metastases [37]. We expect that our model can appropriately disapprove compounds such as Epothilone B during the preclinical animal studies. Brain metastasis of a triple negative breast cancer patient (Appendix A) was used for this experiment. PDXs generated in mouse brain or MFP were passaged three times in the same organ. Four mice each bearing brain or MFP PDXs after third passage were treated with either epothilone B or PBS, and tumor progression was monitored by MRI (brain PDXs) or calipers (MFP PDXs). As we expected, brain PDXs failed to respond to the drug (Figure 6B). Tumor progression was not affected by epothilone B and none of the mice survived beyond 1 month. In contrast, progression of MFP PDXs was retarded by approximately three-fold compared to control mice that received PBS (Figure 6A). Thus, the experiment with PDX-bearing mice replicated the responsiveness to epothilone B that has been observed in humans. Furthermore, we demonstrated that the anti-tumor efficacy of epothilone B is dependent on the tissue that the tumor is in by comparing the drug response between brain and MFP PDXs.

## 3. Discussion

In this study, we developed a novel method to efficiently generate breast cancer brain metastasis orthotopic PDX. The value of our method was illustrated by our finding on engraftment site-specific responsiveness of breast cancer brain metastases to epothilone B that showed that orthotopic PDXs may capture phenotypes that are missed by ectopic PDX as a preclinical model for drug development.

In general, the engraftment rate for PDX generation varies from 4% to 80% [21,38,39]. With our novel method using a pipette with a 10-µL tip, tumors could be prepared and implanted in mouse brain to achieve 100% engraftment and tumor growth without any post-operative mortality. It has been reported that PDX engraftment rates of primary breast cancer vary greatly depending on the subtype [20,40]. In our study, orthotopic PDXs could be equally efficiently generated for both ER+ and ER- breast cancer brain metastases. Moreover, only one cubic mm of tumor was sufficient for engraftment, and tumor growth was observed within almost all mice brains at one month after transplantation. This is important because it demonstrates that our method can be used to generate PDXs relatively quickly without wasting precious human specimens. Although many reports show that PDXs are genetically stable through multiple passages [16,17,18,19], it has also been reported that their genetic phenotype may deviate from the original tumor over time due to selective pressure [41,42]. With our method, passage number can be reduced by transplanting human specimens directly into the mouse brain with a high engraftment rate. The method utilized inexpensive and readily available tools, and was used on more than 40 mice. In contrast, post-operative mortality was high with a tumor implantation method using fine-point forceps, whereas engraftment rate was low with another method using needle for implanting tumors. We believe that in contrast to the other methods, the pipette tip method that we developed allows for more accurate positioning and smoother dispensation of tumor during the implantation procedure. 

In some reports of orthotopic PDXs of brain metastases, needles were used for tumor implantation in the brain and tumors were prepared for implantation by dissociating them enzymatically into a single-cell suspension [43,44]. It has been pointed out that enzymatic digestion of tumors reduces the viability of tumor cells (e.g., [45]). This is in agreement with our study that both engraftment rate and tumor growth rate were significantly reduced by enzymatic digestion. While this could be because of a direct effect of enzymes on cancer cells, the extra time that bulk tumor had to be kept in vitro for enzymatic digestion may have also played a role. Many studies have shown the usefulness of embedding tumors in Matrigel for tumor implantation [22,46]. However, in the current study we observed that the growth of brain PDXs was significantly retarded by Matrigel compared to PBS. We could not find a previous citation of such an observation in the existing literature. The effect that we have observed could be because Matrigel inhibits tumor growth factors in the brain environment. 

Both tumor engraftment and tumor growth rates of PDX tumors are known to increase over time with serial passaging. This has been shown for xenografts of primary breast cancer [18,47] as well as other cancers [47]. However, there has been no report in this regard on the behavior of orthotopic or ectopic PDXs of breast cancer brain metastases. In our study, we found that these PDXs also displayed increased growth rate with time.

As PDX tumors grow, their human tumor stroma is replaced by mouse stroma [48]. In our comparison of transcriptomes of original brain metastasis from one patient and its xenografts in mouse brain and MFP, we observed that approximately 15% of RNA transcripts of PDXs were murine. As expected from the loss of human stromal cells, overall human gene expression in both PDXs was significantly different from the patient’s tumor. Interestingly, our preliminary examination comparing the gene expression at the level of gene pathways implicated that the orthotopic brain PDX was more transcriptionally similar to the patient brain tumor compared to the ectopic MFP PDX. Note that this observation was an analysis made based on single samples from only one patient, thus, it should be regarded as illustration of the potential value of the model. Further study with additional models and biological replicates with more patients is required to validate our proof-of-concept observations finding of transcriptome similarity between patient tumors and their PDXs.

Alzubi et al. demonstrated RNA sequencing data of human breast cancer tumors grown as PDXs at different sites in mouse and separated it into human (cancer) and mouse (microenvironment) transcriptomic datasets. They showed that the cancer transcriptome within PDX was affected by the microenvironment of the tumor implantation site [49]. Rashid et al. utilized a syngeneic immunocompetent mouse model to show transcriptome differences between breast cancer tumors grown in MFP and subcutaneously [50]. These studies show that the tumor microenvironment can affect gene expression in cancer cells of the tumor. Thus, compared to non-brain PDX, transcriptome of the brain PDX is likely to be more similar to the human brain tumor that is implanted. 

A systemic review of 113 phase 3 clinical trials on breast cancer from 2011 to 2017 revealed an overall failure rate of 65% [51]. One of the reasons for this high failure rate is inability to exclude from clinical trials investigational drugs that are not effective in preclinical animal models. Epothilone B is one such drug, whose anti-cancer activity was expected from its taxane-like effect on microtubule stabilization [52]. Though epothilone B has a high anti-cancer activity against primary breast cancer tumors and can cross the brain-blood barrier [36], it failed a phase 2 clinical trial for effectiveness against breast cancer brain metastasis [37]. In our study, while brain metastasis PDXs grown ectopically in mouse MFPs responded to the drug, with tumor growth retarded by about a third compared to untreated tumors, orthotopic PDXs did not respond to treatment. This result suggests that if orthotopic PDXs had been used in preclinical studies of epothilone B, then its inefficiency may have been detected before clinical trials, which could have saved patients from useless therapy and huge expenses as well as emotional turmoil.

Differential responsiveness of brain and non-brain PDXs to anti-cancer drugs has been observed in at least one other study [53]. There are several mechanisms to explain the difference between orthotopic and ectopic PDXs in response to epothilone B. It is possible that the neuronal tumor microenvironment plays a role in reducing sensitivity of orthotopic PDXs to the drug. Indeed, epothilone B is known to have direct effects on neurons and glial cells [54,55]. It is also possible that the brain PDXs showed no response to epothilone B because a sufficiently high drug concentration could not be achieved within tumor tissue. In female nude mice implanted with a colon carcinoma cell-line, epothilone B concentration in tumors growing within brain was found to be approximately a third less compared to tumors growing subcutaneously [36]. With that said, our newly established novel breast cancer brain metastasis PDX model mimicked the clinical trial result, thus, we can expect that our model can appropriately prevent ineffective compounds such as Epothilone B from entering the clinical trial and expose patients with unnecessary side effects when our model is used in preclinical studies. In conclusion, we established and characterized orthotopic breast cancer brain metastasis PDXs that were generated using a novel method that was facile, inexpensive, and efficient. While all animal models have both advantages and limitations, orthotopic PDXs created with our novel method may be powerful tools for preclinical studies of metastatic breast cancer.

## 4. Materials and Methods

### 4.1. Brain Metastases of Human Breast Cancer Patients

All subjects gave their informed consent for inclusion before they participated in the study. The study was conducted in accordance with the Declaration of Helsinki, and the protocol was approved by the institutional review board of Roswell Park Comprehensive Cancer Center (RPCCC; protocol BDR 074516, approved on 9 August 2016). Fresh specimens of brain metastases resected from patients, and associated clinical data were obtained from RPCCC Pathology Research Network. Specimens were aliquoted for formalin fixation and for implantation in mice. Clinical and demographic details of patients whose brain metastases were used in this study are provided in Appendix A. The table also lists the figures for which each patient’s brain metastasis sample was used to generate data.

### 4.2. Experimental Mice

Approval for use of mice was obtained from RPCCC’s Institutional Animal Care and Use Committee. Severely immuno-compromised female mice of strain *NOD-Prkdc^scid^Il2rg*^−/−^ (NSG) and age 6–12 weeks were utilized for tumor implantation. The mice were bred, maintained, and used for experiments at RPCCC’s animal research facility under National Institutes of Health guidelines for the care and use of laboratory animals. Anesthesia was with inhaled isoflurane (3%). Mice also received local subcutaneous injection of buprenorphine (0.05 mg/kg) for pain relief before any surgery. Mice were euthanized with CO_2_ asphyxiation without cervical dislocation.

### 4.3. Preparation of Tumors for Implantation

Fresh tumors from patients’ brains or previously xenografted tumors were transported on ice to the laboratory and prepared for implantation in mice as follows. Tumors were cut into pieces of ~1 mm^3^ volume with a surgical blade. These pieces were directly used for tumor implantation in mouse mammary fat pad. For implantation in mouse brain, the pieces were either used directly or processed further. For some implantations, the pieces were crushed in 2× volume of either Matrigel (Corning^®^, Tewksbury, MA, USA) or phosphate-buffered saline (PBS) with a pipette tip before implantation. For some implantations, the tumor pieces were treated with collagenase (300 U/mL) and hyaluronidase (100 U/mL) from STEMCELL Technologies^®^ (Vancouver, BC, Canada) in RPMI-1640 medium for 1 h at 37 °C, after which the tumor preparation was washed once with centrifugation at 300 g and resuspended to 3 µL with PBS.

### 4.4. Tumor Implantation in Mouse Brain

Anesthetized mice were laid in the prone position and secured with adhesive paper tape. Adequacy of anesthesia was confirmed by checking the mice for absence of toe pinch reflex. An ophthalmic ointment (Puralube^®^, Patterson Veterinary, Greeley, CO, USA) was applied to the corneas of anesthetized mice to prevent eye dehydration. The crown of head was shaved and cleaned with iodopovidone, and a 10-mm incision was made along the midline of the skull. A micro drill (Ideal Micro Drill Burr^TM^, CellPoint Scientific, Gaithersburg, MD, USA) was then used to make a burr hole of 2 mm width 2 mm lateral and 1 mm anterior to the bregma (X, Y, and Z coordinates of 2 mm, 1 mm, and 4 mm with respect to the bregma). Tumor preparation was then injected through the burr hole to a depth of 4 mm in the caudate putamen of brain using one of these three methods. (A) Pipette tip method: the distal 1 cm of a 10-µL polypropylene pipette tip (Tip One^®^, USA Scientific, Ocala, FL, USA) was cut off with scissors, and a 1-mm wide marking was made on the tip at 3 mm distance from the tip’s opening. The tip was then placed. After withdrawing the tip by 1 mm, the tumor preparation (3 µL) was ejected by pushing the pipette plunger. (B) Forceps method: a tumor piece of 1 mm^3^ was picked with fine-point forceps and introduced in the brain. The forceps tip was withdrawn after 1 minute. (C) Needle method: tumor preparation (3 µL) was picked with a 23 G steel needle attached to a 1-mL syringe and introduced in the brain. The burr hole was sealed with bone wax (Ethicon^®^, Somerville, NJ, USA), which is a sterile mixture of beeswax, paraffin, and isopropyl palmitate, for achieving local hemostasis of bone by acting as a mechanical barrier. The scalp incision was closed with a tissue adhesive (Vetbond™, 3M^®^, Maplewood, MN, USA).

### 4.5. Tumor Implantation in Mouse Mammary Fat Pad

A previously established method was followed [30]. Briefly, anesthetized mice were laid in the supine position. A longitudinal midline incision of 5 mm at the level of #2 or #4 mammary gland was made using scissors. The subcutaneous tissue was dissected to visualize the mammary fat pads, which were punctured using fine-point forceps to place one tumor piece of 1 mm^3^ volume at the center of one fat pad. Tumors were implanted in four pads per mouse, except when mice were treated with drugs, in which case only one pad was implanted with a tumor. The midline incision was closed with tissue adhesive.

### 4.6. Tumor Size Measurements

Tumors growing in the mammary fat pad were measured with calipers; tumor volume was estimated from the measurements with formula 1/2(length × width^2^). Tumors in brain were measured by T1/T2 contrast MRI in a 4.7 Tesla scanner with AVANCE electronics and ParaVision software (Bruker^®^, Billerica, MA, USA) at the Translational Imaging shared resource at RPCCC. Briefly, images were processed with Analyze 10.0 software (Biomedical Imaging Resource, Mayo Clinic, Rochester, MN, USA). Regions of interest (ROI) were manually traced with the software’s ROI tool and number of voxels in ROIs were counted. Tumor volume was calculated by multiplying voxel counts with voxel volume.

### 4.7. Tissue Histology

Tissue samples were fixed in 10% buffered neutral formalin and embedded in paraffin for sectioning. Sections of 5 µm thickness were deparaffinized in xylene and rehydrated in an ethanol series. Hematoxylin-eosin staining was performed according to standard protocols. 

### 4.8. RNA Sequencing of Patient-Derived Xenograft Tumor and Original Human Tumor Tissue

The sequencing experiment was performed by the Genomics Shared Resource of RPCCC. Sequencing libraries 1 µg total RNA were generated using RNA HyperPrep kit with RiboErase (KAPA Biosystems^®^, Pleasanton, CA, USA), and sequenced on NovaSeq 6000 instrument (Illumina^®^, San Diego, CA, USA). De-multiplexed sequencing data was processed with Trimmomatic [56] (version 0.35) to remove any adapter and poor-quality sequences. Parameters for Trimmomatic usage were: fa:2:30:10:6:TRUE LEADING:5 TRAILING:5 SLIDINGWINDOW:4:15 MINLEN:36. Resulting read pairs (27.0–29.4 million) were aligned with a HISAT2 [57] aligner (020516 version) against either human GRCh38 or mouse GRCm38 reference genomes (Ensembl version 81) to obtain aligned data (BAM files) with uniquely mapped reads. Alignment rates were 79–83% with the human reference. With the mouse reference, alignment rates were 2% for the brain metastasis sample, and 15% and 13%, respectively, for the mouse brain and mammary fat pad xenografts. The ngs-disambiguate [58] tool (May 2018 version) in Python was used to compare the human and mouse reference-mapped BAM files and generated disambiguated BAMs that had unambiguously human sequences. Gene-level count data were generated from disambiguated BAMs using Subread [59] featureCounts (version 1.5.0-p1) and Ensembl gene annotations (version 81). To generate gene expression values in transcripts per million (TPM) unit, the counts were divided by total exon length of the Ensembl gene identifiers. Human Genome Organization (HUGO) or Mouse Genome Informatics (MGI) gene symbols for the gene identifiers were assigned with Bioconductor package biomaRt (version 2.38.0) [60]; in case of multiple identifiers that had the same symbol, count values were summarized by addition. Data for identifiers without a symbol was removed. For log_2_-transformation, TPM values were padded with 0.05. Unsupervised hierarchical clustering analysis of human gene expression data using the log_2_-transformed TPM values were performed with uncentered Pearson distance and complete linkages. The TPM dataset that was generated in this study is provided in Appendix A.

### 4.9. Analysis of Gene Expression Data

Gene set scores of samples for the Molecular Signatures Database (mSigDB) collection of Reactome gene sets (version 7.0) were calculated with the geneset variation analysis (GSVA) method using GSVA Bioconductor package [61]. Unsupervised hierarchical clustering analysis of the scores and of gene expression data (log_2_-transformed TPM values) were performed with uncentered Pearson distance (cosine distance) and Ward linkages.

### 4.10. Other

Epothilone B was purchased from AdooQ BioScience^®^ (Irvine, CA, USA), and dissolved in 10% v/v dimethyl-sulfoxide in PBS. R and Excel™ (Microsoft^®^, Richmond, WA, USA) software were used for statistical analyses and data plotting. A *p* value threshold of 0.05 was used to deem significance.

## 5. Conclusions

We developed a facile, inexpensive, and efficient method to generate PDXs of human brain metastases of breast cancer in the mouse brain. The method had no perioperative mortality and a 100% engraftment rate. We also compared brain-orthotopic PDXs with ectopic PDXs grown in mammary pads of mice, and show that the responsiveness of PDXs to chemotherapeutic reagents can be dramatically affected by the site that they are in.

## Figures and Tables

**Figure 1 cancers-12-00444-f001:**
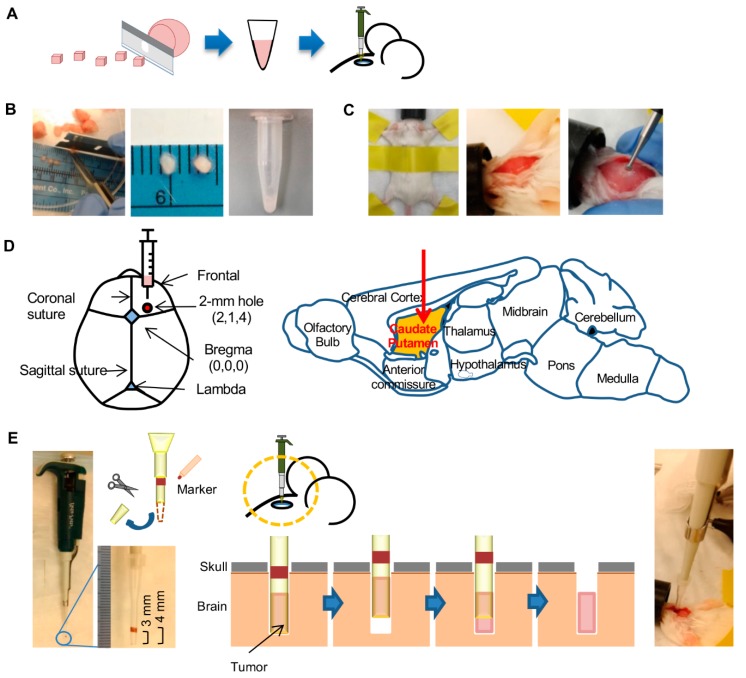
Method for implantation of tumor in mouse brain with the pipette method. (**A**) Workflow of the method involves mincing the tumor into ~1 mm^3^ pieces with a blade. Size of the pieces is then further reduced by crushing the pieces in medium (2× volume) with a pipettor tip before injection of the preparation into mouse brain. (**B**) Photographs shows mincing of a tumor, some minced tumor pieces (ruler markings are in mm), and the injection-ready tumor preparation after crushing of the pieces with a pipette tip. (**C**) Preparation of mouse for tumor injection. An anesthetized mouse is secured in the prone position with tape. The crown of the head is shaved, and a 10-mm incision is made along the midline of the skull. A burr hole of 2 mm width is then made 2 mm lateral and 1 mm anterior to the bregma with a micro drill. (**D**) Coordinates for tumor injection are depicted on anatomic maps of mouse skull and brain. The 2-mm wide burr hole is at X, Y, and Z coordinates of 2 mm, 1 mm, and 4 mm with respect to the bregma. Tumor preparation is injected in the caudate putamen of brain. (**E**) Method of injection of the tumor preparation. The distal 1 cm of a 10-µL pipette tip is cut off with scissors, and a 1-mm wide marking is made on the tip at 3 mm distance from the tip’s opening. Tumor preparation (3 µL) is picked up with the tip attached to a pipette (photograph on left). The tip is then placed through the burr hole (photograph on right) to a depth of 4 mm. After withdrawing the tip by 1 mm, the tumor preparation is ejected by pushing the pipette plunger. The tip is then completely withdrawn, and the burr hole is sealed with bone wax and the scalp is sutured.

**Figure 2 cancers-12-00444-f002:**
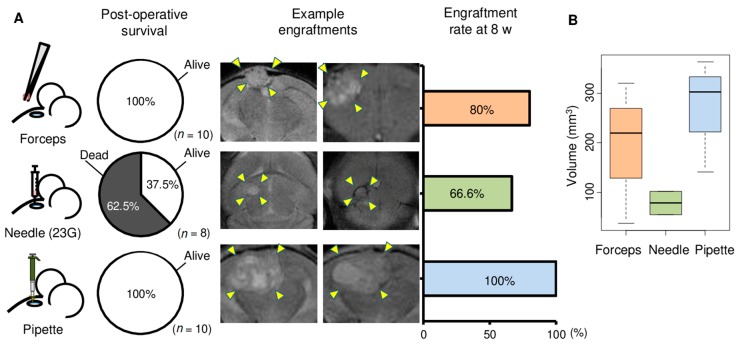
Comparison of three methods for tumor implantation in mouse brain. Brain metastasis of one triple negative breast cancer patient that had been passaged three times as xenograft in mouse brain was used for this experiment. (**A**) Three methods were used for implantation of mouse brain with tumor. The methods used forceps and tissue blocks (*n* = 10), 23 G needle and minced tumor tissue (*n* = 8), or pipette tip and minced tumor tissue (*n* = 10). 1 µL volume of tumor that had been passaged three times as xenograft was used for each mouse. Pie charts depict survival rates within 1 day of implantation with the three methods. Examples of coronal sections of brain in magnetic resonance imaging are shown for two mice each for the three methods. The extent of tumors in the images are indicated with arrowheads. Rates of engraftment of implanted tumor after 8 weeks of surgery for tumor implantation are plotted for the three implantation methods. Mice that did not survive surgery are excluded. (**B**) Tukey boxplots of tumor volumes at 6 weeks in mice with engraftment are shown for the Forceps, Needle, and Pipette methods.

**Figure 3 cancers-12-00444-f003:**
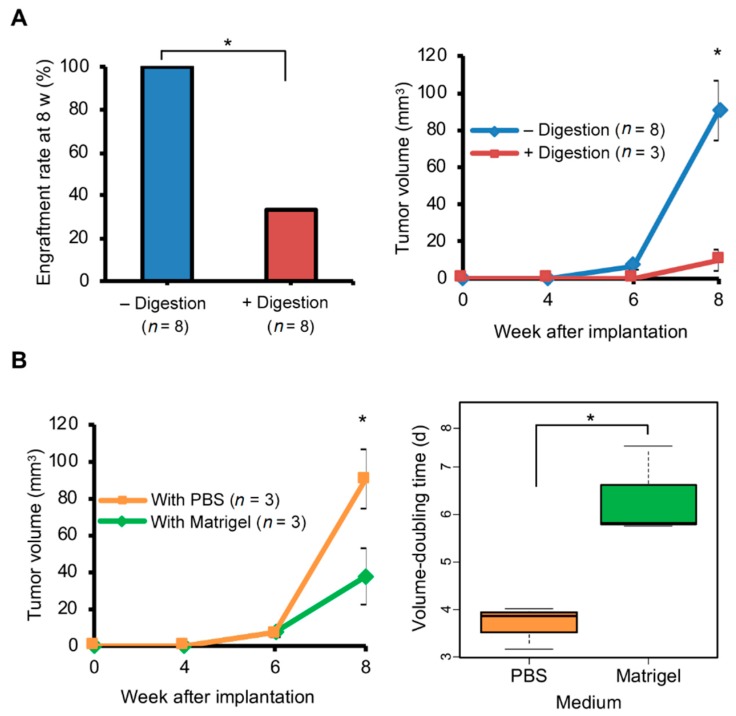
Growth of tumors implanted in the brain is affected by enzymatic digestion and implantation medium. Brain metastasis of one triple negative breast cancer patient that had been passaged three times as xenograft in mouse brain was used for this experiment. All tumor preparations (1 µL tumor with 2 µL phosphate-buffered saline (PBS) or Matrigel) were implanted using the Pipette method, and tumor growth was monitored by magnetic resonance imaging. (**A**) Effect of enzymatic digestion. The engraftment rate at 8 weeks (w) after implantation, and growth over time for engrafted tumors (mean and standard error) are shown for tumors that were either enzymatically digested (+) or not (–) before implantation. PBS was used as medium during implantation. (**B**) Comparison of PBS and Matrigel as medium for tumor implantation. Tumor growth over time for engrafted tumors, and Tukey boxplots of tumor volume-doubling time (days) during weeks 6 and 8 are shown. *p* values in standard *t* tests for group comparison are indicated (* ≤ 0.05).

**Figure 4 cancers-12-00444-f004:**
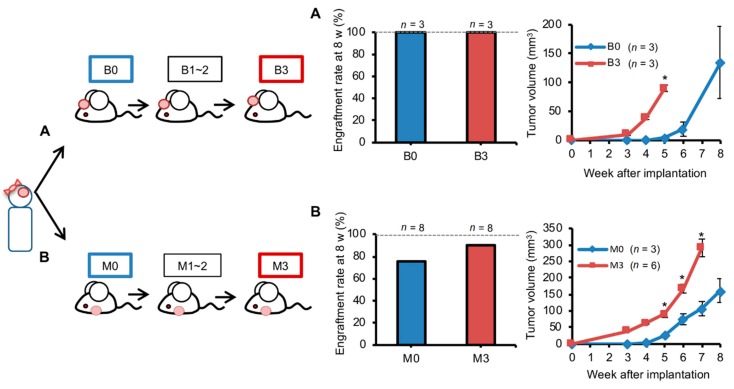
Serial passage facilitates growth of tumors implanted in either brain or mammary fat pad. Brain metastasis of one triple negative breast cancer patient was used for this experiment. Tumor was prepared with the mincing method for implantation (1 µL tumor + 2 µL phosphate-buffered saline) in mouse brain or mammary fat pad with the Pipette or Forceps method, respectively. Implantation was in either brain **(A**) or mammary fat pad (**B**). Implanted tumors were repeatedly passaged three times with the same site used for implantation. Engraftment rate at 8 weeks (w) after implantation, and growth of engrafted tumors over time (mean and standard error) as measured with magnetic resonance imaging (**A**) or calipers (**B**) are shown for tumors before first passage (B0 or M0) and after third passage (B3 or M3). P values in standard *t* tests for group comparison are indicated (* ≤ 0.05).

**Figure 5 cancers-12-00444-f005:**
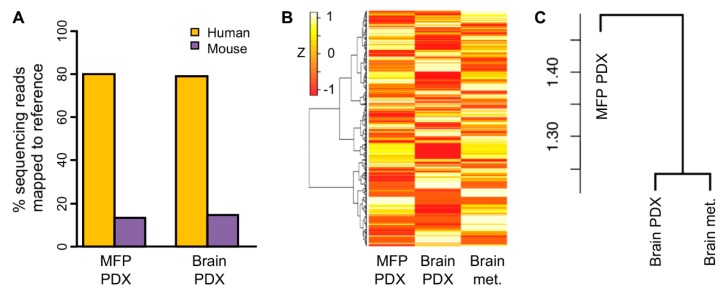
Comparison of gene expression of brain metastasis of breast cancer patient with its implant in mouse brain or mammary fat pad. Brain metastasis of one triple negative breast cancer patient was used for this experiment. Tumor was prepared with the mincing method for implantation (1 µL tumor + 2 µL phosphate-buffered saline) in mouse brain (Brain PDX) or mammary fat pad (MFP PDX) with the Pipette or Forceps method, respectively. Implanted tumors were then passaged three times at the same implantation site before they were harvested for RNA sequencing. (**A**) Mapping of RNA sequencing reads of PDX samples to reference human and mouse transcriptomes. Gene expression of the brain metastasis and implanted tumors was examined by RNA sequencing. (**B**) Heatmap of gene expression is shown for the 25,012 genes identified as expressed among the human and PDX tumors. Gene expression values are Z scaled, and gene clustering is unsupervised. (**C**) Unsupervised hierarchical clustering of gene set variation analysis scores of the three samples for the mSigDb Reactome gene set collection. Dendrogram heights are indicated. For clusterings, cosine distance metric and Ward agglomeration method were used. Sequencing reads that originated from mouse cells in PDX tumor stroma were excluded from the gene expression data to generate the plots in panels B and C.

**Figure 6 cancers-12-00444-f006:**
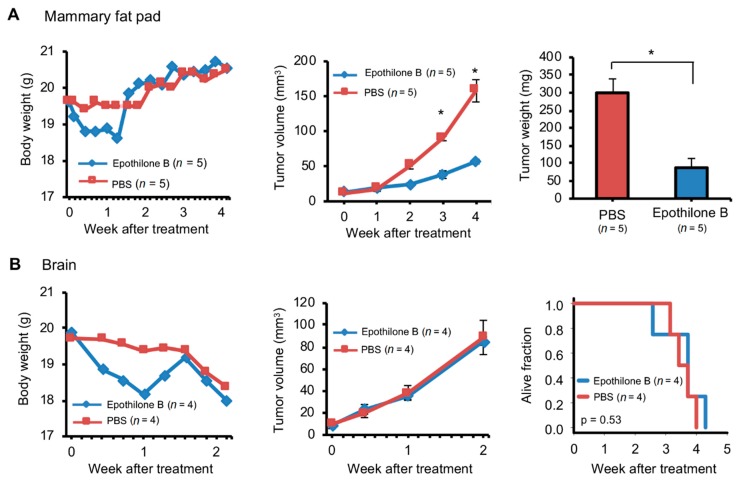
Different responses to chemotherapy of breast cancer patient brain metastasis implanted in the mouse brain or mammary fat pad. Brain metastasis of one triple negative breast cancer patient was used for this experiment. Tumor was prepared with the mincing method for implantation (1 µL tumor + 2 µL phosphate-buffered saline [PBS]) in the mouse mammary fat pad (**A**) or brain (**B**) with the Pipette or Forceps method respectively. Implanted tumors were then passaged three times at the same implantation site. Mice with tumors after third passage were intravenously injected with one dose of epothilone (4 mg/kg) or PBS when tumors were ~10 mm^3^ in volume. Growth of tumors was measured with calipers (**A**) or magnetic resonance imaging (**B**). Body and tumor weights and tumor volumes are plotted (mean and standard error). *p* values in standard *t* tests for group comparison are indicated (* ≤ 0.05). Survival plot for mice with tumors implanted in brain is also shown (*p* value determined with logrank test).

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
