# Peer review of "Novel Breast Cancer Brain Metastasis Patient-Derived Orthotopic Xenograft Model for Preclinical Studies"

_cancers, 2020, doi:10.3390/cancers12020444_

Round 1

Reviewer 1 Report

In the manuscript entitled „Novel breast cancer brain metastasis patient-derived orthotopic xenograft model for preclinical studies” by Oshi et al., authors compare different xenograft models of breast cancer brain metastasis including two different sites of implantation (orthotopic vs. ectopic), three different methods of orthotopic tumor implantation and two tumor pre-implantation treatments (enzymatic digestion and Matrigel plugs).  Additionally, authors analyzed the response to chemotherapy of both orthotopic and ectopic xenografts.

The question posed by the authors is highly interesting, since reliable preclinical models for brain metastatic disease are urgently needed. The experimental work was adequately done and accurately described. The quality of the written English is excellent.

However, this reviewer has two main concerns:

Regarding the orthotopic tumor implantation methods, authors could show that the pipette method leads to a high post-operative survival and a superior engraftment rate after 8 weeks compared with the other two methods. This decisive experiment has been shown only for one breast cancer brain metastasis sample from a triple-negative patient.

Could authors confirm these results with other metastatic tumor samples, maybe from tumors of different molecular subtype? Did they see any difference in the engraftment rate?

Epothilone B has been used in order to address the question of therapy response in PDX models. However, this cytostatic agent is rarely used in the clinic. Why did authors choose particularly this drug?

Further, the lack of response of orthotopic xenografts to Epothilone B has been explained in the discussion part as a possible consequence of tumor microenvironment-induced resistance. What about the role of the blood brain barrier (BBB)in a reduced drug penetration? Is it known whether this substance can cross the BBB? This issue should be additionally addressed in the discussion part.

Reviewer 2 Report

The manuscript proposes an improvement in brain mets PDX generation in murine models and the main takeaway is that their technique results in significantly higher tumor take rates (~100%). 

This reviewer is pleased to see Figure 1, which is very direct and clear in the three different types of tissue/cell implantation into the brain. In Figure 1, E) is not in bold lettering. I imagine that there is some difficulty with "ejecting" the tumor tissue due to its adhesive qualities and tendency to bind to the internal bore of the cut 10uL pipette tip.  Perhaps some actual numbers to indicate this success rate would be helpful? If this does not apply, that is also fine.  Some description of bone wax would be helpful to those who are not DVMs or orthopedic surgeons. 

Figure 2 would have benefited from having the needle cohort in panel B since the engraftment rate at 8wk suggests that there are brain mets/tumors that are quantifiable for panel B. 

If at all possible, H&E sections of these brain mets/tumors in Figure 2 (MRI) would be excellent. 

Figure 3 is very important and in spite of the low N's for some of the cohorts, these results are somewhat consistent with brain PDX experiments. There are reports of higher tumor take rates with digestion (brief expansion in vitro which is a major difference between those works and this submission), but this appears to be more of a person-to-person variation rather than the enzymatic digestion.  Still, the data is convincing and will likely spur others to undertake the same procedure due to the difficulty of accessing the brain and the precious origin of tissue.

There is little mention or clinical detail regarding the origin of the tissue used.  Any description appears to be nebulous and this paper woudl benefit from a table that provided this information. It would emphasize the novelty of brain metastasis tissue being used. 

Reviewer 3 Report

This is an important area of research and an easy paper to read. I think the broader concepts are of interest to the field, however, I am not convinced that the experiments have been performed in a rigorous enough manner; this deficit weakens the strength of the  conclusions. 

It appears that the data in figure 2 is all generated from one PDX, and that this experiment was only performed once, with 8-10 mice used per method. Though there were many replicates within this experiment, there needs to be a replication of data, meaning a second experiment performed to ensure these results hold up over time. It is unclear which PDXs were used in each figure. There should be a chart made which describes all the PDX models used and which figures they were used in. In figure 3, is all this data was from one PDX? It would be informative to know if these results were found across many PDX models. There have been recent papers that have separated the cancer and organ microenvironment with PDX RNA-sequencing data. How does your work fit in with these studies? Has this RNA-sequencing data been deposited into a public database? Were there only 3 samples RNA-sequenced? If there are no biological replicates, it is unknown how strong the conclusions are. The human RNA-sequencing data should be integrated with a large publicly available dataset. This would give us a better understanding of how different these 3 samples are from each other and inform us which intrinsic subtype these samples represent. One of the findings of the paper is that drug treatment responses are different in the brain metastases versus mammary gland tumors, with Epothilone B, working well in the primary tumors, but not the metastases.

(A) Many different drugs could have been selected where this same phenotype would have been observed. What is the rational of treating with Epothilone B? 

(B) The simplest explanation for the lack of efficacy is that the drug in the brain is that it did not cross the blood-brain barrier. Is this drug known to cross this barrier? Was drug concentration in the brain assessed, in either a tumor bearing or non-tumor bearing mouse?

(C) Does each of the 3 methods tested cause a disruption in the BBB?

(D) Lower drug sensitivity in PDX brain metastases as compared to mammary tumors, or other metastases, has been shown with PDXs, some of these data should be referenced.

The discussion usually does not include figure callouts (Figure X). Lines 84-85, 118, English edits needed.

Round 2

Reviewer 1 Report

all questions raised in my first review have been satisfactorily addressed in the revised version and rebuttal letter.

Therefore, I recommend the publication of the revised manuscript in its present form.

Reviewer 3 Report

This revised manuscript has been improved. Written primarily as a methodology paper, I can appreciate the significant in vivo efforts described in these studies, and I do think that readers of this manuscript may gain useful information from figures 1 to 4.

In contrast however, while interesting, there are not enough samples presented in figure 5 to accurately draw scientific conclusions; additional models and biological replicates are needed to support the conclusion that "Transcriptomic profile of orthotopic PDX was more similar to the original brain tumor than ectopic PDX". The approach is good, but it is unknown if this assumption would hold up under more rigorous testing. Furthermore, in the "Response to Reviewers" (Comment 6), the tSNE plot clearly shows that the MFP PDX and Brain PDX are closer to each other than the patient tumor, which is the opposite result presented in the limited analysis in Figure 5. This is by far the major weakness in the paper.

The authors need to check that the references are correctly numbered. It appears that some references were added or removed and I do not see some of them listed in the current manuscript.
